# Roles of Autophagy-Related Genes in the Pathogenesis of Inflammatory Bowel Disease

**DOI:** 10.3390/cells8010077

**Published:** 2019-01-21

**Authors:** Sup Kim, Hyuk Soo Eun, Eun-Kyeong Jo

**Affiliations:** 1Department of Microbiology, Chungnam National University School of Medicine, Daejeon 35015, Korea; mysskks@naver.com; 2Department of Medical Science, Chungnam National University School of Medicine, Daejeon 35015, Korea; 3Infection Control Convergence Research Center, Chungnam National University School of Medicine, Daejeon 35015, Korea; 4Department of Radiation Oncology, Chungnam National University Hospital, 282, Munwha-ro, Jung-gu, Daejeon 34952, Korea; 5Department of Internal Medicine, Chungnam National University Hospital, 282, Munwha-ro, Jung-gu, Daejeon 34952, Korea; liver@kaist.ac.kr; 6Department of Internal Medicine, School of Medicine, Chungnam National University, 266, Munwha-ro, Jung-gu, Daejeon 35015, Korea

**Keywords:** autophagy, ATGs, intestinal homeostasis, inflammatory bowel diseases

## Abstract

Autophagy is an intracellular catabolic process that is essential for a variety of cellular responses. Due to its role in the maintenance of biological homeostasis in conditions of stress, dysregulation or disruption of autophagy may be linked to human diseases such as inflammatory bowel disease (IBD). IBD is a complicated inflammatory colitis disorder; Crohn’s disease and ulcerative colitis are the principal types. Genetic studies have shown the clinical relevance of several autophagy-related genes (ATGs) in the pathogenesis of IBD. Additionally, recent studies using conditional knockout mice have led to a comprehensive understanding of ATGs that affect intestinal inflammation, Paneth cell abnormality and enteric pathogenic infection during colitis. In this review, we discuss the various ATGs involved in macroautophagy and selective autophagy, including *ATG16L1*, *IRGM*, *LRRK2*, *ATG7*, *p62*, *optineurin* and *TFEB* in the maintenance of intestinal homeostasis. Although advances have been made regarding the involvement of ATGs in maintaining intestinal homeostasis, determining the precise contribution of autophagy has remained elusive. Recent efforts based on direct targeting of ATGs and autophagy will further facilitate the development of new therapeutic opportunities for IBD.

## 1. Introduction

Inflammatory bowel disease (IBD) is a complicated autoimmune disorder with multiple etiologies including genetic predisposition, environmental factors and immune-associated pathogenesis [1]. Both Crohn’s disease (CD) and ulcerative colitis (UC), the major clinical phenotypes of IBD, are systemic diseases associated with autoimmune manifestations [2,3]. Although the intestinal host defense is maintained by balancing inflammation and the immune response, excessive inflammation may damage the intestine and its mucosal barrier [1,3]. Although IBD is known to be a polygenic disorder, emerging evidence indicates that genetic susceptibility associated with host autophagy is an important factor in the pathogenesis of IBD [4,5]. 

Autophagy is a cytosolic process that triggers lysosomal degradation of cytosolic materials to maintain intracellular homeostasis under conditions of stress by recycling metabolic building blocks [6,7]. Intracellular cargos sequestered by autophagosomes include damaged cellular organelles, large protein aggregates and intracellular pathogens [8,9]. It is now clear that activation of autophagy contributes to the amelioration of excessive inflammatory responses [10,11]. Dysfunctional or dysregulated autophagy can lead to diverse inflammatory, immune and metabolic disorders [10,11]. Previous studies have demonstrated the involvement of genetic variations of autophagy genes, including *ATG16L1* and *IRGM*, in the pathogenesis of colitis [12,13,14,15,16,17,18]. More recently, it was shown that autophagy-related genes (ATGs), such as optineurin (*OPTN*), transcription factor EB (*TFEB*) and leucine-rich repeat kinase (*LRRK*), are associated with increased susceptibility to colitis, suggesting that these genes are important in colonic immune homeostasis [19,20,21,22,23,24,25].

This review will focus on recent progress in elucidating the roles of ATGs in colonic inflammation and their clinical relevance. We will highlight recent findings regarding several ATGs and the mechanisms through which colitis severity is regulated.

## 2. Overview of Autophagy, Selective Autophagy and ATGs

Macroautophagy (herein referred to as autophagy) is an intracellular catabolic process through which cytoplasmic cargos are sequestered and delivered to lysosomes for degradation [26]. Autophagy plays a critical role in the maintenance of cellular homeostasis during a variety of stress responses, including starvation, hypoxia, toxicity and inflammation [27]. Although it was originally believed that autophagy was a nonspecific process that occurred under starvation conditions, it is now known that autophagy can target specific intracellular organelles or foreign pathogens for timely degradation, which is known as selective autophagy. A detailed discussion of the general aspects of autophagy is beyond the scope of this review; there are numerous excellent reviews dealing in detail about autophagy [28,29]. Here, we briefly review nonselective and selective autophagy, as well as ATGs (Figure 1), before focusing on the relationship of ATGs and autophagy with the pathogenesis of IBD.

### 2.1. Autophagy

Autophagy plays an important housekeeping function in cells through the removal of superfluous or damaged organelles in lysosomes. The autophagic process consists of multiple stages: initiation and biogenesis of autophagosomes, followed by maturation and fusion with lysosomes (Figure 1A) [30]. Autophagy is initiated by the formation of the phagophore, in which the edges of isolation membranes elongate and engulf cytoplasmic cargos [30]. Once the double-membrane structure contains cytoplasmic cargos, autophagosomes undergo maturation to form a completed autophagosome structure and ultimately are fused with a late endosome and lysosome to initiate degradation of cargos [31,32]. 

Regulation of autophagy is important to prevent cell death and pathogenic conditions [33]. It is now clear that autophagic activity is tightly regulated by molecular machinery and transcription factors at transcriptional and post-translational levels [33]. Recent studies have identified several nuclear transcription factors that coordinate autophagy via transcriptional activation of ATGs [33]. For example, *TFEB* plays an essential role in lysosomal biogenesis and activity during autophagy (Figure 2) [34]. We also briefly highlight the involvement of *TFEB* in the regulation of colonic inflammation (Figure 2).

### 2.2. Selective Autophagy

In addition to nonselective degradation, autophagy also plays a role in the targeting and clearance of specific targets/substrates, that is, selective autophagy, which is named depending on its specific targets and includes mitophagy, xenophagy and aggrephagy (Figure 1B) [35,36,37,38]. Selective autophagy involves several steps, including a degradation cue, cargo recognition via selective autophagy receptors, ubiquitination and the recruitment of autophagosome machinery but it does not necessarily occur in a stepwise manner [39,40]. Several ubiquitin binding proteins, such as p62, neighbor of breast cancer 1 (NBR1), OPTN and NDP52/CALCOCO2, have been identified as autophagy receptors responsible for the delivery of ubiquitinated cargos to the autophagy system [41,42,43,44,45,46]. Cargo signals can be classified as ubiquitin-dependent and -independent recognition [40]. Autophagic cargo receptors contain the LC3-interacting region (LIR) motif, which connects cargos to ATG8 family proteins for selective autophagic degradation [45,46]. Despite advances in knowledge of the mechanisms and players involved in canonical and noncanonical autophagy, we still lack a clear understanding of its function in a variety of physiologic and pathologic responses. A few reports have demonstrated the involvement of several autophagic receptors, including p62 and OPTN, in the control of intestinal homeostasis. However, additional autophagic receptors or regulators of autophagic signaling pathways must operate to ameliorate excessive colonic inflammation. Finally, it will be important to investigate the mechanisms by which autophagic receptors or ATGs impact clinical outcomes.

### 2.3. ATGs and the Control of Autophagy

Each step of the autophagy process is highly orchestrated by numerous ATGs; nearly 40 ATGs have been identified in yeast and orthologs of yeast ATGs have been identified in higher eukaryotes with some exceptions, such as mammalian ATG101 [47,48,49]. Among these ATGs, certain gene groups are required for autophagosome formation and are shared among various types of autophagy, such as nonselective and selective autophagy. Mammalian ATGs can be divided into several functional clusters including the ULK1-ATG13-FIP200-ATG101 protein kinase complex, the PtdIns3K class III complex containing VPS34, VPS15 and Beclin 1, the ubiquitin-like ATG5/ATG12 complex and the ubiquitin-like ATG8/LC3 conjugation system [50]. These ATGs participate in different stages of autophagy, such as the induction of autophagosome formation, expansion of phagophores and autophagosome completion [50,51]. 

Recent studies have identified and reported the roles of numerous cargo receptors including NBR1, multi-domain scaffold/adaptor protein p62/sequestosome-1 (p62/SQSTM-1), nuclear domain 10 protein 52 (NDP52) and OPTN [41,42,44]. These selective autophagic receptors contain the LIR motif, thereby connecting ubiquitin-tagged substrates to ATG8 family members such as microtubule-associated protein 1A/1B-light chain 3/γ-aminobutyric acid receptor-associated protein (LC3/GABARAP) [42,45,52].

The roles of ATGs in autophagy have been described in detail in numerous review articles [26,30]. In this review, we focus on ATGs that play essential roles in IBD pathogenesis in terms of autophagy regulation. 

## 3. Overview of IBD

IBD is a disease in which chronic inflammation of intestinal cells occurs due to unknown causes [53]. CD and UC are classified according to the clinical features and characteristics of the disease. Both diseases have similar clinical symptoms including diarrhea, abdominal pain, hematochezia and weight loss; however, the location of inflammation, infiltration degree and complications differ [54]. In general, CD is known to mediate Th1 cell-mediated inflammatory responses and UC is known to mediate Th2 cells [55]. Recently, loss of the suppressive functions of interleukin (IL)-17A-producing regulatory T cells was reported to cause IBD [56]. In general, both types of IBD are treated with anti-inflammatory drugs, such as 5-aminosalicylic acid and corticosteroids; however, in the absence of clinical improvement following treatment with anti-inflammatory drugs, patients achieved a high remission rate using anti-tumor necrosis factor (TNF)-α drugs [57]. However, more than one-third of IBD patients do not respond to anti-TNF-α drugs [58]. Recently, a new therapeutic target for IBD has emerged and the role of Paneth cells in intestinal homeostasis is discussed.

Crypts, concave structures of granulated cells clustered in the base of the small intestine, contain 5–12 Paneth cells. Unlike ordinary enterocytes, which have an average lifespan of 3–5 days, Paneth cells have a longer life expectancy of 20 days [59]. Paneth cells can differentiate into three different cell lineages: enterocytes, goblet cells and enteroendocrine cells [60]. Paneth cells exhibit antimicrobial effects by secreting secretory granules containing antimicrobial peptides (AMPs) and other peptides including lysozymes, alpha-defensins and secretory phospholipase A1 in response to cell stimuli to crypt lumen [61,62,63]. Paneth cell secretion of AMPs plays an important role not only in clearing invading pathogens but also in maintaining the diversity and quantity of intestinal microbiota via intestinal antimicrobial function [64]. Although Paneth cells are normally localized to the small intestine, diseases such as chronic inflammation may result in intestinal metaplasia, which is characterized by the localization and function of Paneth cells in aberrant sites, such as the colon [65]. These metaplastic Paneth cells protect the colonic epithelium from bacterial invasion [59]. However, Paneth cell loss may occur in situations with acute inflammation such as Grade II/III graft versus host disease or in CD [66,67]. In this case, Paneth cells are replaced with lysozyme-producing mucus cells, which can be followed by the development of diseases such as IBD in the small intestine [68]. 

Paneth cells present in the intestinal epithelium are essential for maintaining the homeostasis of normal colonizing microbes of the host. The important pathway in this process is xenophagy, an autophagic pathogen removal process that allows the host to maintain normal metabolic function [69,70]. However, when dysfunction of Paneth cells occurs due to environmental or genetic influences, AMPs are not secreted properly. As a result, dysbiosis, a discrepancy in the composition of normal intestinal microbiota, occurs, which is an important cause of intestinal disorders, especially IBD [71]. For example, in CD patients with impaired xenophagy, adherent-invasive *Escherichia coli* (AIEC) or *Salmonella typhimurium* colonize intestinal epithelial cells (IECs) due to the autophagic dysfunction of Paneth cells [72,73,74]. Thus, the impairment of autophagy in Paneth cells makes it difficult to treat incoming pathogenic bacteria as well as to respond to alterations in the composition of the intestinal microbiota [75]. Ultimately, the poor xenophagy of Paneth cells makes the intestinal epithelium hypersensitive to infiltrating microbes or their products and promotes bacterial self-proliferation and the onset of IBD [76,77].

## 4. ATG Involvement in IBD Pathogenesis

The clinical diversity and heterogeneity of the IBD phenotype are likely due to the presence of genetic heterogeneity together with environmental factors. Susceptibility to IBD may be due to an interaction of several genes, identified by genome-wide association studies (GWASs) [78,79,80]. To date, over 200 loci have been identified as genetically significant loci by a meta-analysis combined with GWAS [81,82]. Earlier independent GWASs showed that autophagy gene variants, including autophagy-related gene 16-like 1 (*ATG16L1*) and immunity-related GTPase M (*IRGM*) are linked to CD susceptibility highlighting the role of autophagy in controlling infection, inflammation and cancer [13,15,83,84]. Furthermore, gene mutation or deletion studies have indicated that the autophagy pathway affects the onset and exacerbation of IBD via several mechanisms including clearance of invading bacteria, secretion of granules from Paneth cells, inflammasome activity, pro-inflammatory cytokine production and endoplasmic reticulum (ER) stress. However, the role of autophagy in the pathogenesis of IBD is still debated. Although many researchers focused on the involvement of ATGs in IBD pathogenesis, little is known about the autophagic role of ATGs and the mechanisms that confer intestinal inflammation. Table 1 summarizes the ATGs and transcription factors described in this review and their functional relationships in intestinal pathogenesis. 

### 4.1. ATG16L1

Numerous studies have reported that genetic variation in *ATG16L1* is associated with IBD risk in ethnically diverse populations [13,14,92,93,94,95,96]. Notably, the rs2241880 single nucleotide polymorphism (SNP; T300A) of *ATG16L1* was repeatedly found in several Caucasian cohorts, suggesting a strong association of this variant with the incidence of CD, although it was not frequently found in other populations, particularly in Asian patients [13,84,96]. 

ATG16L1, a homolog of ATG16, is essential in the formation of autophagosomes, along with the ATG12-ATG5 conjugate [97,98]. Importantly, Cadwell et al. showed that mice with low expression of *ATG16L1* (ATG16L1^HM^ mice) exhibited abnormal Paneth cell granule secretion and that mice with *ATG16L1* deficiency in Paneth cells had a defect in the granule exocytosis pathway [76]. Similarly, patients carrying the *ATG16L1* risk allele (T300A) had pathological features such as disorganized granules or diffuse Paneth cell cytoplasmic lysozyme staining [76]. Using IEC-specific *ATG16L1*-deficient mice and ex vivo IEC organoids, a recent study showed that *ATG16L1* in IECs played an essential role in controlling pathology, intestinal inflammation and TNF-induced apoptosis [86]. Additionally, previous studies showed that the ER stress sensor inositol-requiring enzyme (IRE)-1α accumulated in Paneth cells of *ATG16L1*^ΔIEC^ mice and CD patients (T300A), suggesting that defective autophagy leads to pathological activation of IRE1α to drive intestinal inflammation [85]. Moreover, loss of IKKα function markedly impaired the secretion of cytoprotective IL-18 and upregulated ER stress responses through decreased ATG16L1 stabilization [99]. These data emphasize the role of ER stress in defective *ATG16L1*-mediated colonic inflammation [85,99]. Indeed, IEC-specific deletion of *ATG16L1* or *ATG7* led to hyper-activated ER stress, which may amplify the severity of intestinal inflammation in autophagy-defective conditions [100]. 

Although IECs, particularly Paneth cells, are important in defective *ATG16L1*-associated intestinal inflammation, the function of ATG16L1 in myeloid cells has also been demonstrated [87]. Saitoh et al. showed that *ATG16L1*-deficient macrophages exhibited Toll/IL-1 receptor domain-containing adaptor inducing interferon (IFN)-β (TRIF)-dependent activation of the inflammasome, resulting in the production of high amounts of the inflammatory cytokines IL-1β and IL-18 [87]. Deficiency of *ATG16L1* in hematopoietic cells resulted in an increased susceptibility to dextran sulfate sodium (DSS)-induced colitis, suggesting an essential role for ATG16L1 in the control of intestinal inflammation [87]. Another study using mice with myeloid *ATG16L1* deficiency showed exacerbated colitis with upregulated proinflammatory responses as well as increased colitogenic bacteria, indicating that *ATG16L1* deficiency results in alterations in macrophage function that affect the severity of CD [88]. 

Peripheral blood mononuclear cells (PBMCs) isolated from CD patients with the *ATG16L1* T300A risk variant have been shown to exhibit increased production of the proinflammatory cytokines IL-1β and IL-6, particularly in response to NOD2 ligands [101]. Moreover, the loss of *ATG16L1* increased TRIF and its signaling, resulting in increased production of type I IFN and IL-1β [102]. Interestingly, the genetic variant *ATG16L1* T300A was found to be associated with adalimumab treatment, suggesting that this SNP affects the response to treatment with immunomodulatory drugs [103]. Importantly, the CD risk allele T300A variant (T316A in mice) is associated with accelerated degradation of ATG16L1 due to caspase-3 activation. Upon apoptotic stimuli or metabolic stress, human and murine macrophages harboring T300A or T316A variants of *ATG16L1*, respectively, exhibited accelerated degradation of ATG16L1, leading to decreased autophagy, defective clearance of the pathogen and enhanced inflammation [104]. These data strongly suggest that the functional defect in ATG16L1 is involved in the dysregulation of intestinal homeostasis and CD pathogenesis (Figure 3).

### 4.2. IRGM

Human immunity-related guanosine triphosphatase (GTPase) family M (IRGM) encodes the only functional immunity-related GTPase (IRG) among IRG family members [105]. Involvement of genetic polymorphisms of *IRGM* in CD and tuberculosis has been previously demonstrated, particularly in GWASs [16,106,107]. A meta-analysis showed that the *IRGM* variants rs13361189 and rs4958847 are associated with both UC and CD in human IBD [12]. However, another study in a Korean population showed that selected SNPs of *IRGM* were associated with CD but not UC susceptibility [17]. 

Human *IRGM* (syn: *LRG47*, *IFI1*), which is encoded by the immunity-related GTPase protein family, M gene (*IRGM*; 5q33.1), is thought to be distant from a class of IRGs in mice. There are more than 20 IRG genes (*IRGM1–3*, *IRGN1–8*, *IRGB1–10* and *IRGD*) in mice, whereas there is only one *IRGM* gene present in humans and chimpanzees, making the study of the role of IRGs *in vivo* difficult. Earlier studies showed that the murine GTPase IRGM1 (LRG-47) was important for autophagy activation to eliminate intracellular *Mycobacterium tuberculosis* [108,109,110]. A further study showed that a human IRG protein, the human ortholog IRGM1 (IRGM), contributed to the control of *M. tuberculosis* through autophagy activation [111]. Tiwari et al. reported an essential function of Irgm1 as an innate effector in targeting the mycobacterial phagosome through lipid-mediated binding to enhance phagosome maturation and the antimicrobial response [112]. Furthermore, IRGM has been shown to regulate autophagy by translocating to the mitochondria and influencing mitochondrial fission, which is required for autophagic defense against intracellular mycobacteria [113]. Studies using *IRG*-deficient mice showed that IRGM can be induced by IFN-γ and plays a role in the clearance of intracellular bacteria including *Toxoplasma gondii*, *Listeria monocytogenes* and *Salmonella* spp. as well as mycobacteria (Figure 4) [110,114,115,116,117,118]. Human IRGM and murine IRGM1 contribute to cell-autonomous defense though autophagy activation via the recruitment of both autophagic and SNARE adaptor proteins during infection (Figure 4) [18,110,111,117,119,120,121,122]. However, IRGM favors viral replication through autophagy activation. For example, IRGM is translocated to the Golgi apparatus, where it regulates Golgi membrane fragmentation and is involved in virus-triggered autophagy activation during hepatitis C infection [123].

Murine and human studies have demonstrated the protective role of IRGM in the maintenance of intestinal homeostasis. *Irgm1*-deficient mice have been shown to exhibit functional defects in intestinal Paneth cells and hyperinflammation in the colon and ileum following chemical exposure [124]. In addition, the IRGM protein contributed to the limitation of CD-associated intracellular AIEC in epithelial cells through autophagy activation and phagosomal maturation [121]. In the intestinal mucosa, greater quantities of pathogenic AIEC, which invade IECs and induce TNF-α, are found in CD patients than in healthy controls [72,125]. These data collectively suggest the importance of IRGM in CD pathogenesis via limitation of pathogenic bacteria through autophagy activation. A recent cohort study revealed the relationship among autophagy-related *IRGM* variants, visceral adipose tissue and nonalcoholic fatty liver disease, which shows an increased morbidity with CD [126]. However, there is still a debate regarding the relevance of autophagy in CD in terms of IRGM, as autophagy activation has been observed in Paneth cells in CD patients, independently of *IRGM* variants associated with CD susceptibility [127]. Moreover, RNA analysis showed that most autophagy gene sets were downregulated by appendectomy, which contributed to protection against UC [128]. Suppression of autophagy may offer cross-reactive immunity between host antigens and microbes through decreased antigen processing, thereby ameliorating symptoms of colitis [128].

### 4.3. LRRK2/MUC19

LRRK2/MUC19 is a complex protein that contains a RAS of complex proteins (ROC) GTPase domain, a C-terminal ROC domain and a Ser/Thr kinase domain and is involved in NOD2-mediated signaling, of which autophagy is a downstream process [129]. Because *LRRK2* is a well-known gene involved in the pathogenesis of Parkinson’s disease (PD), most earlier studies were performed in neuronal cells [130,131]. Later, meta-GWASs identified the links between *LRRK2* and CD and leprosy, suggesting a role for LRRK2 in immune regulation during infection and inflammation [80,132,133]. LRRK2 is highly expressed in myeloid cells and B cells which is induced by IFN-γ and is involved in the production of inflammatory cytokines and antimicrobial responses in macrophages [80,132,133]. In addition, LRRK2 is required for commensal bacteria-driven cargo sorting through recruitment to lysozyme-containing dense core vesicles in Paneth cells, thereby participating in the coordination of the lysozyme-sorting process in the intestine to promote symbiosis [134].

A previous genome-wide linkage analysis suggested that a locus on chromosome 12 (historically known as PARK8) is linked to familial parkinsonism in the Japanese population [135]. Further studies have demonstrated the involvement of *LRRK2* in autosomal-dominant parkinsonism in multiple families [130,131]. In addition, two meta-GWASs reported *LRRK2* as a CD susceptibility gene [79,80]. A genome-wide conjunctional analysis revealed several novel loci, which are potentially involved in the association between PD and autoimmune diseases [136]. For example, known PD loci adjacent to *LRRK2* (rs17467164) were proposed as overlapping susceptibility loci for UC and CD [136]. Accumulating data in conjunction with the development of *in silico* analyses may identify novel genetic variants that affect the risk of several diseases occurring in combinatorial patterns. In a Japanese CD cohort, a defective Paneth cell phenotype was correlated with clinical characteristics and autophagy-associated *LRRK2* (*LRRK2*M2397T) was associated with Paneth cell defects [22]. The majority of *LRRK2* SNPs, which are associated with IBD, are found in non-coding intronic regions [24]. There is speculation that there might be a relationship between the high frequency of non-coding region SNPs in *LRRK2* and the stability/expression levels of LRRK2 [24]. 

Several studies have investigated the mechanistic aspects of pathogenic LRRK2 in PD models (Figure 5). Pathogenic LRRK2 is involved in protein translation through regulation of microRNA function (let-7 and miR-184*), which results in altered production of E2F1/DP and is critical for cell cycle and survival [137]. The autosomal dominant mutant protein LRRK2 phosphorylates and activates transcription of the forkhead box transcription factor FoxO1, which is crucial in the upregulation of cell death molecules and is associated with LRRK2-mediated cell death [138]. Importantly, *LRRK2* deficiency led to impairment of the autophagy-lysosomal pathway, altered expression of LC3-II and p62 and increased α-synuclein aggregates in the kidney, in an age-dependent manner [139]. Recent studies have shown that mitochondrial RHOT1-dependent mitophagy is delayed with the PD mutant *LRRK2*G2019S, suggesting a critical function of LRRK2 in the regulation of mitophagy [140]. Although LRRK2 is known to be involved in autophagic flux, the exact roles and mechanisms by which LRRK2 controls intestinal homeostasis are not completely understood in terms of autophagy regulation [129,139]. 

The molecular mechanisms underlying how LRRK2 affects the pathogenesis of CD have not been widely examined. An earlier study showed that *LRRK2* deficiency led to increased susceptibility to DSS-induced colitis in mouse models by negatively regulating activation of the transcription factor NFAT [23]. In a recent study, both lymphoblastoid cells from control patients bearing a high-risk allele of *LRRK2* and dendritic cells from CD patients exhibited elevated LRRK2 expression, which resulted in severe colitis with increased Dectin-1-mediated NF-κB activation and proinflammatory cytokine responses [25]. As an IFN-γ target gene, LRRK2 induction and function were investigated in immune cells [133,141]. LRRK2 was previously detected in inflamed intestinal tissues, particularly in macrophages of the lamina propria and was shown to play a role in host defense through regulation of reactive oxygen species generation [133]. These studies collectively suggest that the fine-tuning of LRRK2 is required for the prevention and treatment of colitis and related infections (Figure 5).

### 4.4. ATG7

ATG7 is an E1-like activating enzyme that facilitates autophagosome formation through two ubiquitin-like conjugation systems, LC3 lipidation and Atg12 conjugation. Availability of ATG conditional deletion mice have improved our understanding of the contribution of different ATGs in specific cells/tissues and provided insight into the role of individual ATGs in intestinal homeostasis during colitis. In intestinal cells, the function of Atg7 has been studied using intestinal epithelium-specific (tamoxifen-inducible) *Atg7* knockout (*ATG7*^IEC-KO^) mice. An earlier study showed that *ATG7*^IEC-KO^ mice had a similar pathology in the ileum and Paneth cell abnormality with defective granule exocytosis as those observed in Atg16L1HM and Atg5flox/floxvillin-Cre mice [142]. Consistent with these observations, *ATG7*^IEC-KO^ mice exhibited decreased granule size and decreased lysozyme levels in Paneth cells and increased production of TNF-α and IL-1β mRNA in response to lipopolysaccharide in the epithelium of the small intestine when compared to those of control small intestinal tissue [143,144]. A further study showed that the exacerbated experimental colitis in *ATG7*^IEC-KO^ mice was associated with abnormal microflora composition and dysregulated expression of antimicrobial or antiparasitic peptides (angiogenin-4, Relmβ, intelectin-1 and intelectin-2), as well as suppressed secretion of colonic mucins [145]. 

*ATG7* conditional knockout mice also exhibited increased susceptibility to and reduced clearance of *Citrobacter rodentium* infection in the intestinal epithelium during *C. rodentium* infectious colitis [89]. A recent study emphasized the role of autophagy in controlling intestinal homeostasis using mice with conditionally deleted *ATG7* in CD11c+ antigen-presenting cells (*ATG7*^ΔAPC^), which enhanced immunopathology and inflammatory Th17 responses, as well as abnormal mitochondrial function and oxidative stress [90]. Another group showed that mice with myeloid cell-specific deletion of *ATG7* exhibited increased susceptibility to experimental colitis accompanied with increased colonic inflammation [91]. Furthermore, *ATG7* deletion in intestinal epithelium-specific *XBP1*-deficient mice synergistically aggravated the intestinal pathology, resulting in the development of extensive submucosal or transmural inflammation and recapitulated the features of human CD, suggesting that autophagy contributes to a compensatory process in the intestinal epithelium during sustained ER stress [100]. These data strongly indicate that *ATG7* is crucial for controlling intestinal inflammatory responses and defense against the virulence of enteric pathogens to maintain intestinal homeostasis (Figure 2). However, there is little evidence that ATG7 is clinically relevant in IBD.

## 5. Selective Autophagic Receptors and IBD

The involvement of the autophagic receptors p62 and TFEB in IBD pathogenesis has been reported [21,146]. Several studies have also identified SNPs of *NDP52* and *OPTN* in individuals with IBD [20,147,148]. Here, we discuss the current evidence regarding the role of these cargo receptors in terms of IBD.

### 5.1. p62

The ubiquitin-binding protein sequestosome 1 (SQSTM1/p62) is a well-known autophagy adaptor and was initially identified as a p56lck binding protein [149]. Mutations of *p62* are known to be associated with Paget’s disease of bone [150]. An earlier study showed that a direct interaction between p62 and LC3/GABARAP family members led to autophagy-mediated destruction of p62-positive, polyubiquitin-containing bodies [151]. In addition, Komatsu et al. showed the function of p62 in selective autophagy activation via the binding of ubiquitinylated protein aggregates for delivery to LC3 autophagosomes [152]. In canonical autophagy, accumulation of p62 in the cytoplasm is generally regarded as a sign of reduced autophagic activity and the impaired autophagy, because increased autophagic flux degrades p62 (Figure 2) [153]. Overall, p62 is an essential scaffold protein that can bind a variety of partner proteins, participating in diverse biological signaling that affects innate immunity, apoptosis, inflammatory responses and tumorigenesis [154].

Although there are a few reports on the involvement of p62 in the pathogenesis of colitis, defective autophagy with decreased turnover of p62 levels has been observed in intestinal inflammation [155,156]. In a study performed in an epithelial cell line, the intracellular survival of AIEC LF82 bacteria, which promote the gastrointestinal inflammatory response, was higher in cells silenced for *p62* than in cells transduced with empty vector (ShCTR) [157]. In animal and human studies, defective autophagic flux with elevated p62 was observed in IBD tissues and models [156]. Furthermore, immunohistochemical expression of p62 was higher in epithelial cells of damaged mucosa than in those of non-damaged mucosa [146]. Understanding how p62 regulates intestinal homeostasis will enable the development of more effective therapeutic strategies against IBD.

### 5.2. Optineurin (OPTN)

OPTN is a selective autophagy adaptor protein that plays an important role in mitophagy and xenophagy. OPTN is involved in various biological responses including vesicular trafficking, anti-bacterial and antiviral responses and autophagy and interacts with numerous cellular proteins including myosin VI, Rab8 and Tank-binding kinase 1 [158,159,160]. The clinical relevance of OPTN has been indicated by the genetic variants/mutations of OPTN linked to glaucoma, Paget’s disease of bone and amyotrophic lateral sclerosis [161,162,163]. OPTN plays an essential role in mitophagy and is involved in several neurodegenerative diseases, including PD [160,164]. OPTN is also required for the clearance of pathogens, such as *Salmonella* and *Listeria*, to promote activation of xenophagy [43,165]. 

A few studies have identified *OPTN* deficiency in CD patients [19,20]. The function of OPTN in the regulation of intestinal homeostasis was suggested in IRE1α-driven colitis as an IRE1α-interacting protein [85]. *OPTN* deficiency led to an accumulation of IRE1α, which enhanced colitis pathology during ER stress (Figure 2) [85]. Indeed, a subgroup of CD patients with low OPTN expression has been identified [20]. The same study showed that *OPTN* deficiency led to decreased production of TNFα and IL-6 and increased susceptibility to Citrobacter colitis and *E. coli* peritonitis (Figure 2) [19]. Thus, these data suggest the importance of OPTN in macrophage inflammation and bactericidal function to promote the antimicrobial host defense and may explain the link between *OPTN* deficiency and increased CD pathogenesis [19]. These insights indicate that selective autophagy and cargo receptors may play important roles in the regulation of colonic homeostasis (Figure 2). A better understanding of the players and mechanisms underlying selective autophagy in intestinal inflammation will aid in the discovery of new therapeutic targets for IBD.

## 6. Transcription Factor *TFEB*

TFEB, a member of the microphthalmia-associated transcription factor (MITF)/transcriptional factor E (TFE) family, has been identified as a key regulator of autophagy maturation and lysosome biogenesis [166,167,168,169]. TFEB activation is required for clearance of pathogenic molecular aggregates in neurodegenerative diseases, such as α-synuclein and aberrant tau protein, to promote therapeutic effects in PD and Alzheimer’s disease, respectively [170,171,172]. 

Our understanding of the effects of TFEB in IBD is currently in its infancy. A recent study showed that mice with a conditional deletion of *TFEB* in the intestinal epithelium (*TFEB*^ΔIEC^) had a defect in Paneth cell granules, lower expression levels of lipoprotein ApoA1 and exaggerated colitis upon DSS injury [21]. Further studies evaluating the function of TFEB and its clinical relevance in IBD pathogenesis are needed, given the essential role of TFEB in regulating the autophagy lysosome pathway [168]. Elucidation of the involvement of TFEB and other transcription factors in colitis will improve our understanding of the mechanism of autophagic regulation in the complicated pathogenesis of IBD. 

## 7. Conclusions

Studies over the last decade have suggested that genetic variants of several ATGs are highly associated with IBD susceptibility. Although the distinct genetic variations or manipulations provide strong support for the causative role of ATGs in the pathogenesis of IBD, they also raise two important questions: whether defective autophagy is a major trigger for pathogenic inflammation in IBD and what signaling pathways control canonical and noncanonical autophagy in intestinal epithelial cells/tissues. We focused our review on ATGs, selective autophagy receptors and transcription factors involved in maintaining intestinal homeostasis in human and mouse studies.

Both ATG16L1 and IRGM, two important ATGs in IBD, have principally been investigated in human and murine intestinal epithelial cells, particularly in Paneth cells, the major secretory cells of the small intestine. Recent studies have suggested that defective ATG16L1-mediated inflammation is due to aberrant ER stress, as upregulated IRE1α was observed in Paneth cells from *ATG16L1*^ΔIEC^ mice and CD patients (T300A). The interactions between ATGs and other biological systems such as the ER stress response may have a significant impact on the pathogenesis of IBD as well as other inflammatory diseases. Furthermore, genetic association studies have suggested that *LRRK2/MUC19* and *ATG7* deficiency aggravate intestinal inflammation in a mouse model of colitis. 

Selective autophagy receptors, p62 and OPTN and the transcription factor TFEB have been suggested to play key roles in controlling intestinal inflammation and homeostasis. Future studies to further define the mechanisms by which cargo receptors contribute to specific types of autophagy will enhance our understanding of intestinal homeostasis in terms of autophagy regulation. This knowledge will ultimately aid in the development of novel therapeutic strategies and drug targets for combating intractable chronic inflammatory diseases such as IBD.

## Figures and Tables

**Figure 1 cells-08-00077-f001:**
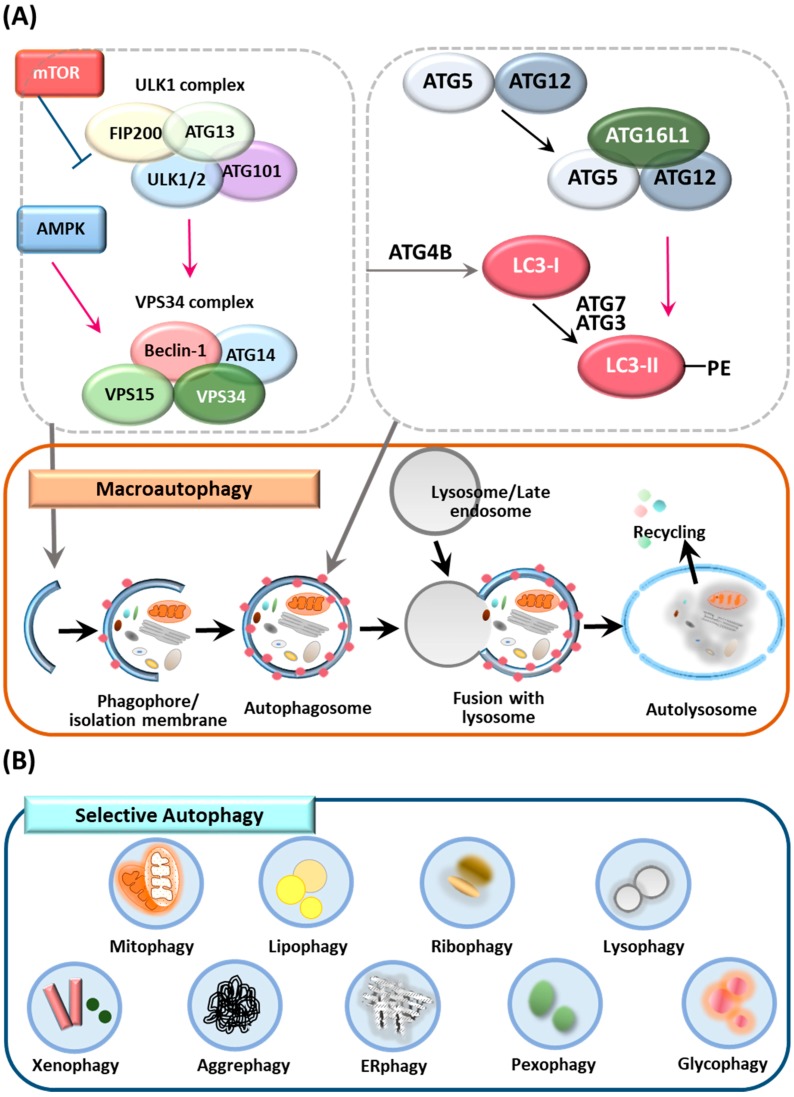
Overview of autophagy, selective autophagy and ATGs (**A**) Molecular machinery of autophagy process. After mTOR inhibition or AMPK activation, the autophagy process begins with the biogenesis of the phagophore/isolation membrane. The ATG16L1-ATG5-ATG12 and LC3-II-PE conjugates participate in autophagosome formation process. The mature autophagosomes are fused with a late endosome and lysosome to initiate degradation of cargos. Finally, cells recycle the released products in cytosol. (**B**) Selective autophagy clears various targets such as subcellular structure, bacteria, protein and lipid aggregates.

**Figure 2 cells-08-00077-f002:**
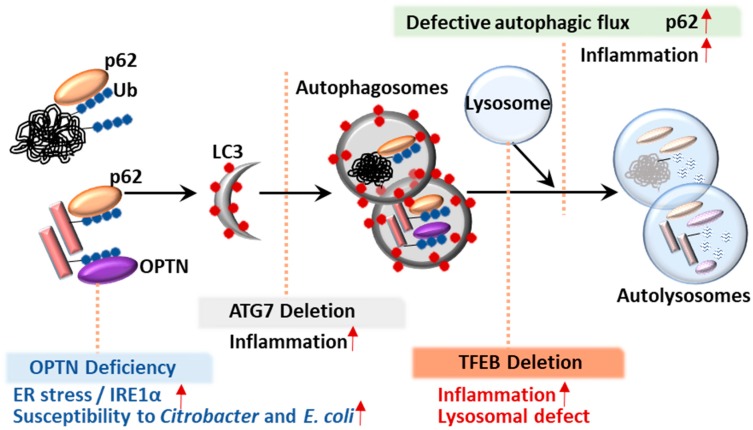
The process of autophagy flux and involved genes. The autophagy flux is depicted. A normal autophagic flux includes the autophagosome formation and maturation step and the autolysosome formation step. The possible conditions associated with involved genes are depicted: (1) *OPTN* deficiency leads to an accumulation of IRE1α and increased susceptibility of *Citrobacter* and *E. coli*. (2) *ATG7* deletion is associated with increased inflammation. (3) *TFEB* deletion results in increased inflammation and lysosomal defect.

**Figure 3 cells-08-00077-f003:**
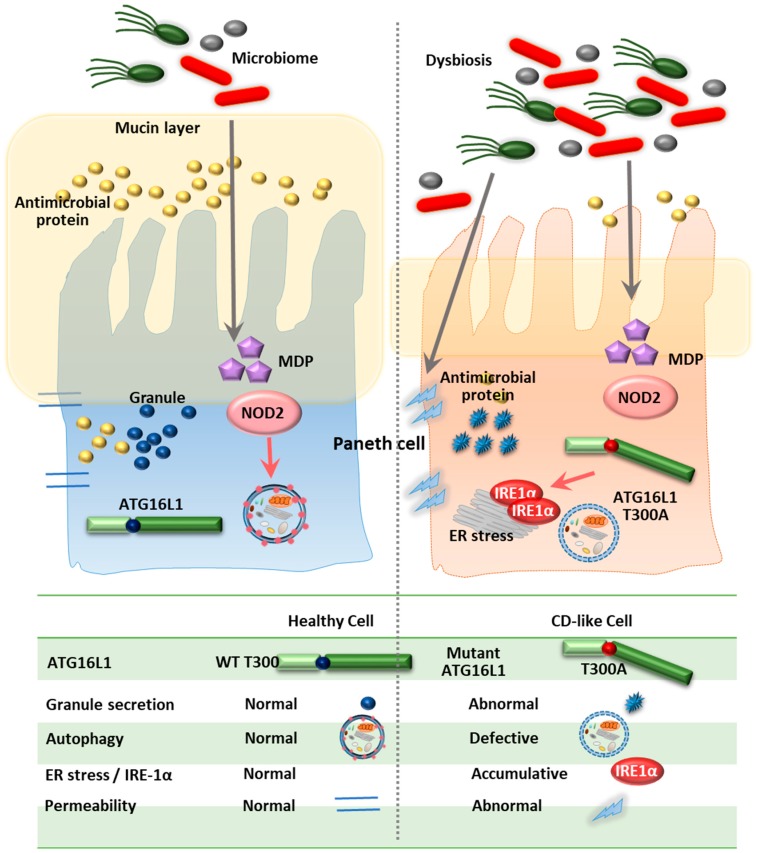
Summary diagram showing the role of ATG16L1 in the Crohn’s disease. The left panel demonstrates the normal host defense mechanism against intracellular pathogens. Healthy cells exhibit normal granule secretion, autophagic activity, ER stress response and permeability. The right panel shows the ATG16L1 T300A variant cells defective in granule secretion, autophagy process, IRE1α degradation and tight junction barrier function.

**Figure 4 cells-08-00077-f004:**
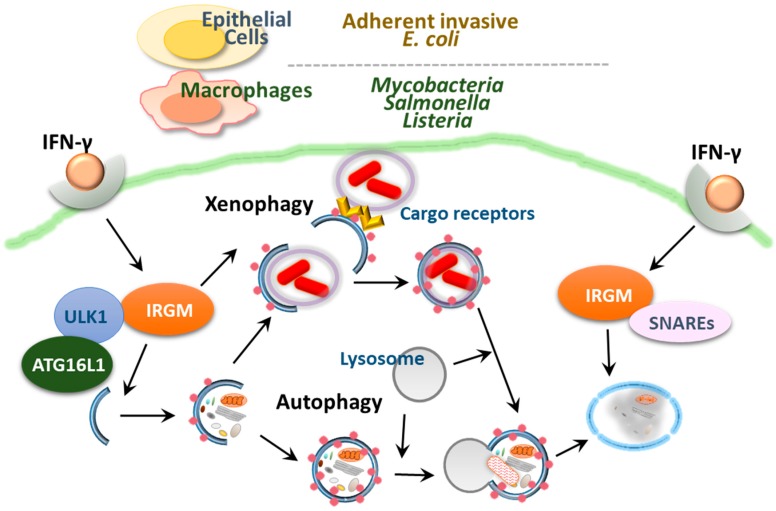
Autophagy targets adherent-invasive *Escherichia coli* (AIEC), Mycobacteria, Salmonella and Listeria by different mechanisms. Stimulation with IFN-γ induce IRGM to clear intracellular bacteria. Furthermore, IRGM can be induced by IFN-γ contribute to cell-autonomous defense though autophagy activation via the recruitment of both autophagic and SNARE adaptor proteins during infection.

**Figure 5 cells-08-00077-f005:**
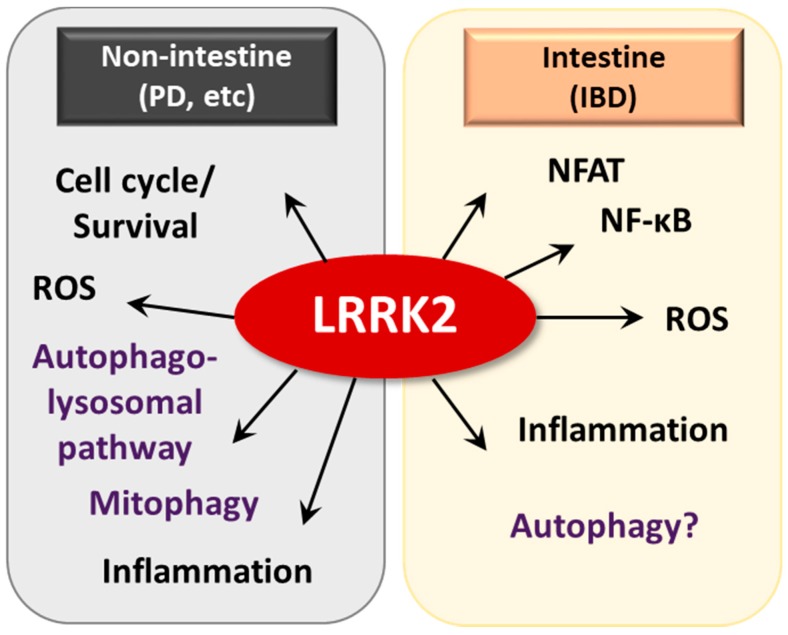
The pathogenic roles of LRRK in non-intestine and intestine diseases. In Parkinson’s disease, LRRK2 is involved in cell cycle/survival, ROS generation, autophagolysosomal pathway, mitophagy and inflammation. LRRK2 plays a key role in intestine homeostasis through regulating NFAT, NF-κB, ROS generation, inflammation and autophagy.

**Table 1 cells-08-00077-t001:** Genetic models related to autophagy in intestinal pathogenesis.

Knocked-Out Gene	Cell Type	Mechanism	Outcome	Reference
***ATG16L1***	Intestinal epithelial cells	Abnormality of Paneth cell granule secretion and defect in the granule exocytosis pathway		[76]
	Intestinal epithelial cells	ER stress sensor inositol-requiring enzyme (IRE)-1α accumulated in Paneth cells	Increased intestinal inflammation	[85]
	Intestinal epithelial cells	Elevated pro-inflammatory cytokine secretion and increased IEC apoptosis after *Helicobacter hepaticus* infection	Exacerbated murine model of chronic colitis	[86]
	Myeloid cells	Production high amounts of the inflammatory cytokines IL-1β and IL-18 via Toll/IL-1 receptor domain-containing adaptor inducing interferon (IFN)-β (TRIF)-dependent activation of the inflammasome	Increased susceptibility to dextran sulfate sodium (DSS)-induced colitis	[87]
	Myeloid cells	Increased reactive oxygen species production, impaired mitophagy, reduced microbial killing, impaired processing of MHC class II Ags and altered intracellular trafficking to the lysosomal compartments	Exacerbated murine model of acute and chronic colitis	[88]
	Myeloid cells		No effect on disease severity in murine model of chronic colitis	[86]
***IRGM***	Intestinal epithelial cells	Marked alterations of Paneth cell location and granule morphology	Hyperinflammation in the colon and ileum following chemical exposure	[85]
***LRRK2/*** ***MUC19***	Myeloid cells	Activation of the transcription factor NFAT	Increased susceptibility to DSS-induced colitis in mouse models	[23]
***ATG7***	Intestinal epithelial cells	Higher expression levels of pro-inflammatory cytokine mRNA in the large intestine after infection	Increased susceptibility to *Citrobacter rodentium* infectious colitis in mouse models	[89]
	Intestinal antigen presenting cells	Enhanced immunopathology and inflammatory Th17 responses, as well as abnormal mitochondrial function and oxidative stress	Increased susceptibility to DSS-induced colitis in mouse models	[90]
	Myeloid cells	Increased colonic cytokine expression, T helper 1 skewing and systemic bacterial invasion	Increased susceptibility to DSS-induced colitis in mouse models	[91]
***OPTN***	Myeloid cells	Decreased antimicrobial host defense (decreased production of TNFα and IL-6) after infection	Increased susceptibility to *Citrobacter* colitis and *E. coli* peritonitis	[19]
***TFEB***	Intestinal epithelial cells	Defect in Paneth cell granules, lower expression levels of lipoprotein ApoA1	Exaggerated colitis upon DSS injury	[21]

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
