# Peer review of "Roles of Autophagy-Related Genes in the Pathogenesis of Inflammatory Bowel Disease"

_cells, 2019, doi:10.3390/cells8010077_

Round 1

Reviewer 1 Report

The Review contribution is well organized and covers the state-of art of the autophagy involvement within bowel disease. Main actor genes are reported along with their normal or pathogenic functions. Figures are well designed and clear to the audience.

The only request I can require, deriving from the general title used (i.e. intestinal homeostasis), is an additional paragraph of few summarized lines, describing literature evidences on the involvement of autophagy-related genes in other diseases such as celiac and other inflammatory/autoimmune where intestinal homeostasis are largely compromised.

Please, correct some type-spacing mistakes 

Author Response

The Review contribution is well organized and covers the state-of art of the autophagy involvement within bowel disease. Main actor genes are reported along with their normal or pathogenic functions. Figures are well designed and clear to the audience.

The only request I can require, deriving from the general title used (i.e. intestinal homeostasis), is an additional paragraph of few summarized lines, describing literature evidences on the involvement of autophagy-related genes in other diseases such as celiac and other inflammatory/autoimmune where intestinal homeostasis are largely compromised.

Thank you for your helpful suggestions. We have changed the title of the paper to “Roles of Autophagy-related Genes in the Pathogenesis of Inflammatory Bowel Disease” to focus on IBD.

Please, correct some type-spacing mistakes 

We agree and have fixed these errors throughout the paper.

Reviewer 2 Report

This is a comprehensive and well written review of autophagy in intestinal homeostasis. The figures are of high quality.

The first paragraph focuses on IBD although the title of the manuscript does not include IBD. I advise to write first about normal intestinal homeostasis and then about diseases. Alternatively, the title may be changed to focus on IBD.

Page 2, line 51: The statement "autophagy can generate energy by recycling metabolic building blocks" requires a reference or should be modified. Does autophagy generate energy?

Table 1 states that a mutation of p62/Sqstm1 increased the intracellular survival of bacteria. However, the associated reference 156 reports a study on the shRNA-mediated knockdown of p62. It is misleading to speak of a "mutation".

Author Response

This is a comprehensive and well written review of autophagy in intestinal homeostasis. The figures are of high quality.

The first paragraph focuses on IBD although the title of the manuscript does not include IBD. I advise to write first about normal intestinal homeostasis and then about diseases. Alternatively, the title may be changed to focus on IBD.

Thank you for your helpful comments. We have changed the title of the paper to “Roles of Autophagy-related Genes in the Pathogenesis of Inflammatory Bowel Disease” to focus on IBD.

Page 2, line 51: The statement "autophagy can generate energy by recycling metabolic building blocks" requires a reference or should be modified. Does autophagy generate energy?

Correct. This sentence has been removed.

Table 1 states that a mutation of p62/Sqstm1 increased the intracellular survival of bacteria. However, the associated reference 156 reports a study on the shRNA-mediated knockdown of p62. It is misleading to speak of a "mutation".

  We agree. In table 1, p62 section has been removed, since it might cause a confusion. Furthermore, Table 1 has been newly organized into several parts, such as “Genetic variants”, “Types of variants”, “Proposed phenotype” to improve readers’ understanding.   .